# Physically and Chemically Stable Anion Exchange Membranes with Hydrogen-Bond Induced Ion Conducting Channels

**DOI:** 10.3390/polym14224920

**Published:** 2022-11-15

**Authors:** Chengpeng Wei, Weisheng Yu, Liang Wu, Xiaolin Ge, Tongwen Xu

**Affiliations:** Anhui Engineering Laboratory of Functional Membrane Materials and Technology, Collaborative Innovation Centre of Chemistry for Energy Materials, School of Chemistry and Material Science, University of Science and Technology of China, Hefei 230026, China

**Keywords:** anion exchange membranes, hydrogen bonding interaction, ion transport channels, alkaline stability, fuel cells

## Abstract

Anion exchange membranes (AEMs) with desirable properties are the crucial components for numerous energy devices such as AEM fuel cells (AEMFCs), AEM water electrolyzers (AEMWEs), etc. However, the lack of suitable AEMs severely limits the performance of devices. Here, a series of physically and chemically stable AEMs have been prepared by the reaction between the alkyl bromine terminal ether-bond-free aryl backbone and the urea group-containing crosslinker. Morphology analyses confirm that the hydrogen bonding interaction between urea groups is capable of driving the ammonium cations to aggregate and further form continuous ion-conducting channels. Therefore, the resultant AEM demonstrates remarkable OH^−^ conductivity (59.1 mS cm^−1^ at 30 °C and 122.9 mS cm^−1^ at 90 °C) despite a moderate IEC (1.77 mmol g^−1^). Simultaneously, due to the adoption of ether-bond-free aryl backbone and alkylene chain-modified trimethylammonium cation, the AEM possesses excellent alkaline stability (87.3% IEC retention after soaking in 1 M NaOH for 1080 h). Moreover, the prepared AEM shows desirable mechanical properties (tensile stress > 25 MPa) and dimensional stability (SR = 20.3% at 90 °C) contributed by the covalent-bond and hydrogen-bond crosslinking network structures. Moreover, the resulting AEM reaches a peak power density of 555 mW cm^−2^ in an alkaline H_2_/O_2_ single fuel cell at 70 °C without back pressure. This rational structural design presented here provides inspiration for the development of high-performance AEMs, which are crucial for membrane technologies.

## 1. Introduction

Anion exchange membranes (AEMs) have drawn great interest from researchers because of their essential role in a wide range of electrochemical devices, including anion exchange membrane fuel cells (AEMFCs) [1,2], redox flow batteries [3] and water electrolyzers [4]. The items determine the performance of these electrochemical devices directly [5]. Although numerous AEMs have been designed over the years, till now, ideal AEMs with high ion conductivity and excellent chemical stability are still in urgent need, which has been hindering the realization of high-performance electrochemical devices [6]. There are two main challenges for the development of AEMs: (i) many AEMs are chemically unstable and tend to degrade via many pathways (such as nucleophilic substitution and Hofmann degradation) under alkaline conditions, leading to the intractable service life of AEMs [7,8]; (ii) most AEMs possess unsatisfactory ion conductivity due to the intrinsic low diffusion coefficient of hydroxide ions [9].

ATMs typically consist of polymer backbones and cation groups, and therefore, the AEMs stability is under the influence of the two components. Although various polymer backbones (including poly(arylene ether), polyolefin, polyphenylene, poly(norbornene), and so on) [10,11,12,13] and cation groups (e.g., ammonium, imidazolium, piperidinium, organometallic cations and phosphonium) [14,15,16,17,18] have been developed for AEMs, most of AEMs show insufficient tolerance to an alkaline environment. Years of study have shown that the poly(arylene ether) backbones are alkali-sensitive and peculiarly prone to degrade via aryl-ether bond cleavage [19]; by contrast, the aryl ether-free backbones display outstanding alkaline stability [20,21]. Typically, Bae and co-workers first synthesized the aryl ether-free polymer backbone for AEMs via the superacid catalyzed polycondensation of electron-rich phenyl monomers and trifluoromethyl ketones, and the resultant AEMs show excellent alkaline stability in 1 M NaOH at 80 °C [22]. Recently, Lee and co-workers reported that AEM composes of a quaternized ether-free polycarbazole backbone [23]. The AEM is tolerant to the chemical attack of OH^−^, and the hydroxide conductivity and ion exchange capability (IEC) are almost unchanged even after immersing in 1 M KOH at 80 °C for 1000 h.

Moreover, the durability of AEMs is also largely determined by the cation groups. The benzyl trimethylammonium cation (BTMA) is the most commonly used cationic group for AEMs because of its simple preparation. Unfortunately, research shows that BTMA is vulnerable to the attack of OH^−^ and degrades fairly rapidly via Hoffmann elimination and nucleophilic substitution, thus leading to cation loss [24]. Encouragingly, Tomoi’s a group reported that the alkylene chain-modified trimethylammonium cation shows improved alkaline stability compared to BTMA since the absence of reactive benzylic carbon attached directly to the quaternary nitrogen [25]. Hibbs’s group and Coughlin’s group further replaced the benzylic methylene spacer of BTMA with a hexamethylene spacer, and the resultant hexane-6-trimethylammonium cation is significantly stable under strongly alkaline conditions [26,27]. Therefore, the remarkably stable aryl ether-free backbone and alkylene chain-modified trimethylammonium cation become the ideal candidate for preparing long-term stable AEMs.

Due to the tradeoff between ionic conductivity and water uptake, it remains a challenge to effectively improve the ion conductivity of AEMs [28]. The ion conductivity typically increases with the IEC, while high IEC generally results in excess water uptake, which in turn leads to excess membrane swelling and poor mechanical strength [29]. Although the detrimental tradeoff can be mitigated by regulating the AEM architectures [30], this method shows limited effectiveness. The rational design of AEM structures to drive the cation groups to aggregate and form continuous ion channels is believed to be an efficient way to achieve fast ion transport [31]. Numerous studies have revealed that interconnected ion channels can be constructed by optimizing polymer architectures. Typically, for the traditional side-chain type AEMs, the polarity contrast between hydrophobic polymer backbones and hydrophilic cations can drive the hydrophilic cations to aggregate and to facilitate the formation of ion conducting channels [32,33]. Furthermore, multiblock and densely grafted AEMs were designed with the expectation of constructing ion -conducting channels more effectively by strengthening the polarity contrast of polymer segments [34,35]. However, the driving force brought by the polarity discrepancy is relatively insufficient, limiting the formation of well-connected ion channels. Therefore, to promote the formation of interconnected ion channels, researchers have been devoted to facilitating the good aggregation of cations by introducing an additional strong driving force.

Supramolecular interactions are widely used to construct various assemblies efficiently, such as DNA molecules and phospholipid bilayer [36,37], which possess a significant function in natural biological systems. Inspired by this, various supramolecular interactions, including π-π stacking interaction, ion-dipole interaction, and host-guest interaction, have been introduced to construct desirable ion channel structures in AEMs. For example, Wei et al. reported the AEMs containing quaternary ammonium phthalocyanine macrocycles, and the π-π stacking interaction between the macrocycles results in the formation of fast hydroxide transport channels [38]. Moreover, Xu et al. demonstrated the effectiveness of the ion-dipole interaction between piperidinium cations and dipolar ethylene oxide spacers in forming ion channels [39]. Very recently, Xu et al. successfully constructed ordered ion channels in AEMs relying on the host-guest interaction between crown ether and dibenzylammonium salt [40]. However, this strategy is greatly limited by choice of polymer backbones and cation groups. Most of the reported AEMs with supramolecular interactions induced ion channels are based on the alkali-sensitive structures such as poly(arylene ether) backbone and BTMA cation mentioned above due to the simplicity of their chemical modification. Consequently, the alkaline stability of the AEMs is still unsatisfactory, further limiting the service life of the corresponding devices.

Despite the advancements achieved in optimizing AEMs’ performance, only a few AEMs demonstrated desired comprehensive properties for practical application. Till now, the development of suitable AEMs for various electrochemical technologies remains a significant challenge, as the ideal AEMs must possess high ion conductivity, excellent alkaline stability, and restricted swelling in water simultaneously. Therefore, to ensure the performance and long-term service life of the AEMs-based electrochemical devices, reasonable design, and preparation of AEMs with suitable properties are of great necessity.

In materials science, hydrogen bonding interaction is an ideal supramolecular interaction for constructing advanced assemblies due to its highly directional and selective nature [41]. In addition, the strength of hydrogen bonding interaction is tunable since there are many types of hydrogen bonding (generally including single, double, triple or quadruple hydrogen bonding) [42]. Therefore, hydrogen bonding interaction is promising for constructing well-ordered ion channels in AEMs due to its fascinating characteristics. Combining these advances, we herein present a strategy to achieve AEMs with high ion conductivity and high alkaline stability. As shown in Figure 1, The targeted AEMs were synthesized by the Menshutkin reaction between the alkyl bromine terminal aryl ether-free backbone and urea group-containing crosslinker. Such a structural design brings many benefits to the resultant AEMs. Firstly, the hydrogen bonding interaction between urea groups drives the cations to be aggregated, thereby facilitating the formation of continuous ion -conducting channels. Such ion channel structures are expected to increase the ion conductivity of AEMs. Secondly, the adopted aryl ether-free backbone and alkylene chain-modified trimethylammonium cation are both tolerant to alkaline environments, guaranteeing the alkaline stability of the prepared AEMs. Moreover, the resultant AEMs possess both covalent-bond and hydrogen-bond crosslinking networks, thus endowing the AEMs with excellent dimensional stability. The morphology, ion conductivity, water uptake, alkaline stability and AEMFCs performance of the AEMs were studied systematically to reveal the structure-property relationships. The results reveal that the prepared AEMs demonstrate great practical application potential in membrane technology, and the presented strategy may benefit the development of high-performance AEMs.

## 2. Materials and Methods

### 2.1. Materials

Fluorene, 1,6-dibromohexane, tetrabutylammonium iodide, *p*-terphenyl, 1,1,1-trifluoroacetone, trifluoromethanesulfonic acid and 1,3-bis(3-(dimethylamino)propyl)urea were purchased from Energy Chemical Co., Ltd. (Shanghai, China). Sodium hydroxide, N-methyl pyrrolidone (NMP), chloroform, n-hexane, dichloromethane, ethanol, and diethyl ether were purchased from Sinopharm Chemical Reagent Co., Ltd. (Shanghai, China). All reagents and solvents were used as received.

### 2.2. Synthesis of 9,9-Bis(6-Bromohexyl)Fluorene (BBF)

9,9-bis(6-bromohexyl)fluorene (BBF) was synthesized by the reaction of fluorene and 1,6-dibromohexane [43]. The mixture of fluorene (4.00 g, 24.1 mmol), aqueous sodium hydroxide solution (50 wt%, 80.0 g), tetrabutylammonium iodide (1.70 g, 4.60 mmol) and 1,6-dibromohexane (35.2 g, 144 mmol) was stirred under argon atmosphere at 70 °C for 12 h. After the reaction was completed, the mixture was poured into water and extracted with chloroform. The organic extract was washed with brine, and the solvent was removed by a rotary evaporator. The residue was purified by flash column chromatography (n-hexane as the eluent) to afford BBF as a colorless oil. The ^1^H NMR spectrum of BBF is shown in Figure 2a.

### 2.3. Synthesis of poly(9,9-Bis(6-Bromohexyl)Fluorene-co-p-Terphenyl 1,1,1-Trifluoroacetone) (PBTT-x) Copolymers

The poly(9,9-bis(6-bromohexyl)fluorene-co-*p*-terphenyl 1,1,1-trifluoroacetone) (PBTT-x, where x is the molar ratio of 9,9-bis(6-bromohexyl)fluorene in the copolymer) was synthesized by the superacid-catalyzed condensation reaction [44]. Take the synthesis of PBTT-0.6 as an example. A mixture of BBF (1.60 g, 3.25 mmol), *p*-terphenyl (4.98 g, 2.17 mmol) and 1,1,1-trifluoroacetone (0.73 g, 6.50 mmol) in dichloromethane (10 mL) was cooled to 0 °C, and then trifluoromethanesulfonic acid (8.0 mL) was added dropwise. The reaction mixture was then stirred at room temperature for 50 min. The resultant viscous solution was poured into ethanol, and the resulting fibrous solid precipitation was collected and washed with ethanol. Finally, the solid was dried to afford PBTT-0.6 as a white fiber. The ^1^H NMR spectrum of PBTT-x is shown in Figure 2b.

### 2.4. Preparation of the Quaternary Ammonium-Functionalized Crosslinked poly(1,6-Diphenylpyrene-co-p-Terphenyl 1,1,1-Trifluoroacetone-co-Piperidinium) (CPBTT-x) Membranes

The quaternary ammonium-functionalized crosslinked poly(1,6-diphenylpyrene-co-*p*-terphenyl 1,1,1-trifluoroacetone-co-piperidinium) (CPBTT-x) membrane was prepared by the Menshutkin reaction between PBTT-x and 1,3-bis(3-(dimethylamino)propyl)urea. Take the preparation of the CPBTT-0.6 membrane as an example. PBTT-0.6 (1.0 g, 2.50 mmol -CH_2_Br) was dissolved in NMP (35 mL), and then 1,3-bis(3-(dimethylamino)propyl)urea (0.29 g, 2.50 mmol tertiary amine groups) was added. The mixture was stirred at 90 °C for 12 h. After the reaction was finished, the resulting viscous solution was cast on a glass plate and heated at 90 °C for 12 h to afford the targeted CPBTT-0.6 membrane. The CPBTT-0.6 membrane in OH^−^ form was prepared after immersing in 1 M NaOH for 24 h.

### 2.5. Characterizations

^1^H NMR spectra were collected on a Bruker 510 400 MHz instrument. CDCl_3_ was used as the solvent, with TMS as the internal standard. Fourier transform infrared spectroscopy (FTIR) spectra were recorded on an attenuated total reflectance Fourier transform infrared (Nicolet is 10). Field-emission scanning electron microscopy (SEM) (GeminiSEM 500, Oberkochen, Germany) was used to characterize the surface and cross-section morphologies of membranes. Atomic force microscopy (AFM) (DI MultiMode V, Veeco) was used to reveal the surface morphologies of the membranes. Transmission electron microscope (TEM) images were collected on a JEM 2100F field emission transmission electron microscopy (JEOL Ltd., Akishima) with an accelerating voltage of 200 kV. Mechanical properties were recorded on a discovery DMA 800 dynamic mechanical analyzer (TA Instruments). A TGA Q5000 thermogravimetric analysis (TA Instruments) was employed to reveal the thermal stability of membranes. An 850e multi -range fuel cell test station (Hephas Energy Co.) was used to evaluate the single -cell performance of the membrane.

The experimental details for measuring the IEC value, water uptake (WU), swelling ratio (SR), OH^−^ conductivity and alkaline stability of membranes were shown in supporting information. The membrane electrode assembly fabrication for H_2_/O_2_ single -cell test was given in supporting information too.

## 3. Results and Discussion

### 3.1. Polymer Synthesis, Characterization and AEMs Preparation

As illustrated in Figure 1, the targeted AEMs were prepared by a three-step synthesis procedure. The 9,9-bis(6-bromohexyl)fluorene (BBF) monomer was first synthesized from fluorene and 1,6-dibromohexane. The chemical structure of the BBF monomer was confirmed by ^1^H NMR spectroscopy (Figure 2a). Then, the aryl ether-free polymer backbone (PBTT-x) was synthesized via the superacid-catalyzed condensation of BBF, *p*-terphenyl and 1,1,1-trifluoroacetone. The ratio of BBF and *p*-terphenyl was controlled to prepare AEMs with different IEC. In the ^1^H NMR spectrum of PBTT-x, as shown in Figure 2b, the characteristic signals at 3.25 ppm and 2.02 ppm correspond to the -CH_2_Br and methyl groups, respectively. In addition, the signals that appear at 7.18–7.68 ppm and 0.61–1.86 ppm are attributed to the phenyl groups and side-chain methylene groups, indicating the successful synthesis of the polymer backbone. The PBTT-x backbone further reacted with 1,3-bis(3-(dimethylamino)propyl)urea to synthesize the targeted AEMs (CPBTT-x). As the reaction progresses, the reaction solution becomes more and more viscous, suggesting the success of the Menshutkin reaction. Because the CPBTT-x was insoluble in common solvents after crosslinking, the chemical structure of CPBTT-x was verified by FTIR spectroscopy. As shown in Figure 2c, the peak at 2964 cm^−1^ can be assigned to the stretching vibration of -CH_3_ [45]. The absorption peaks at 2926 cm^−1^ and 2855 cm^−1^ are attributed to the symmetric and asymmetric stretching of -CH_2_- in the membrane [46]. The peak at 1656 cm^−1^ and 1564 cm^−1^ are attributed to the stretching vibration of C=O and bending vibrations of N-H in urea groups [47]. In addition, the absorption peak at 1264 cm^−1^ is assigned to the stretching vibration of C-N in quaternary ammonium groups [48], confirming the successful preparation of the targeted AEMs. The practical IEC values of the prepared CPBTT-x AEMs were determined by Mohr titration. The IEC values of CPBTT-0.4, CPBTT-0.5, CPBTT-0.6 are 1.48, 1.63, 1.77 mmol g^−1^, respectively (Appendix A). Remarkably, the practical IEC values are significantly lower than the theoretical IEC values calculated based on the feeding ratio, which is attributed to the incompletion of the crosslinking reaction.

The hydrogen bonding interaction between urea groups was characterized via FTIR spectroscopy [49]. Figure 2d shows the temperature-dependent FTIR spectra of the CPBTT-0.6 membrane upon heating from 30 to 130 °C. The strong C=O stretching vibration absorption band appears at 1678 cm^−1^ at 30 °C while shifting to 1686 cm^−1^ at 130 °C. In addition, the N-H bending band also undergoes a significant shift from 1560 to 1550 cm^−1^ with the temperature increasing from 30 to 130 °C [50]. In the meantime, the intensity of the two characteristic peaks diminishes progressively during the heating process. Due to the intrinsic thermo-responsiveness of the urea-based hydrogen bonding interaction, the ordered hydrogen bonding tends to break and convert to a disordered state upon heating, thus leading to the peak shift and decrease of peak intensity [51]. This result verified the presence of hydrogen bonding interaction in the resulting AEMs, which is expected to promote the aggregation of cation groups and the formation of continuous ion channels.

### 3.2. Morphology Analysis

Appendix A shows the surface and cross-section SEM images of the CPBTT-0.6 membrane. Both the surface and cross-section are homogeneous and devoid of cracks or holes, suggesting the good membrane-forming ability of the crosslinked copolymer. The thickness of the CPBTT-0.6 membrane is approximately 40 nm, as measured according to the cross-sectional SEM image.

AFM phase images (Figure 3a–c) were further performed to understand the influence of hydrogen bonding interaction on membrane morphologies [52]. In the AFM phase image, the bright yellow regions represent the hydrophobic domains consisting of the polymer backbone, while the dark regions correspond to the hydrophilic domains containing quaternary ammonium cations and bound water in AEMs [17]. Figure 3a shows the AFM phase image of CPBTT-0.4. The hydrophilic domains (dark regions) in CPBTT-0.4 are randomly distributed, suggesting the limited self-assembly ability of quaternary ammonium cations. By contrast, for CPBTT-0.5 (Figure 3b) and CPBTT-0.6 (Figure 3c), with the increase of urea groups, the stronger hydrogen bonding interaction promotes the aggregation of cation groups more efficiently. As a result, the CPBTT-0.6 displays the clearest and aggregated hydrophilic domains, and the hydrophilic domains are well connected with each other to form continuous ion transport channels. Figure 3d–f shows the TEM images of CPBTT-x membranes, and similar morphologies were also observed. Compared to CPBTT-0.4 and CPBTT-0.5, the dark regions composed of iodide-stained hydrophilic ionic clusters in the CPBTT-0.6 membrane overlapped to form interconnected ion channels, which are expected to accelerate the ion conduction [53]. The morphological analysis indicates that the hydrogen bonding interaction between urea groups drove the quaternary ammonium cations to aggregate and thus construct connected ion channels in AEMs.

### 3.3. Physical Properties

Water uptake is an important issue of AEMs because water is a benefit for promoting the hydration of ammonium cations and dissociation of OH^−^, which eventually lead to high ion conductivity. However, excess water absorption tends to lead to undesirable dimensional stability and insufficient mechanical strength, limiting the practical applications of AEMs. Therefore, ideal AEMs should possess suitable water uptake. Figure 4a,b show the WU and SR of the prepared CPBTT-x AEMs. Both WU and SR of the AEMs increase accordingly owing to the increase of IEC or temperature, in line with expectations. As shown in Figure 4c, the water contact angles of the CPBTT-x membranes follow the trend of CPBTT-0.4 > CPBTT-0.5 > CPBTT-0.6, suggesting IEC-dependent water uptake behavior. The CPBTT-0.6 membrane with the highest IEC shows moderate WU of 80.2% at 30 °C and 99.3% at 90 °C, respectively. At the same time, the CPBTT-0.6 membrane shows a slight change in SR as temperature increases. The SR of CPBTT-0.6 only increases from 17.6 % to 20.3 % with the increase of temperature from 30 °C to 90 °C, despite the presence of hydrophilic urea groups [54]. The covalent-bond and hydrogen-bond crosslinking structures of CPBTT-0.6 avoid excessive membrane swelling, thereby leading to excellent dimensional stability. The low dimensional change of CPBTT-0.6 is expected to stabilize the membrane electrode assembly (MEA), which is directly related to the operational stability of AEMFCs [55].

In addition, the promising AEMs must possess good mechanical strength and thermal stability for application in AEMFCs. The mechanical strength of CPBTT-x was measured via tensile test, and the corresponding stress-strain curves were shown in Figure 4d. The combination of covalent-bond and hydrogen-bond crosslinking structures endows all the AEMs with good tensile strength (tensile stress > 25 MPa), which meets the requirement of AEMFCs’ application. Moreover, the dynamic and reversible hydrogen bonding can serve as sacrificial bonds for stress dissipation; thereby, all the AEMs possess excellent toughness (elongation at break > 16%). The thermal stability of the CPBTT-x AEMs was further tested by thermogravimetric analysis (TGA), and the resulting TGA and differential thermal gravity (DTG) curves were illustrated in Figure 4e and Appendix A, respectively. The CPBTT-x AEMs show three notable weight-loss stages during the heating process. The first weight-loss stage occurs between 210 and 350 °C which is mainly associated with the degradation of quaternary ammonium cations. The second stage, between 420 and 500 °C, probably results from the degradation of urea groups [47]. The third stage, which starts above 500 °C, corresponds to the decomposition of the polymer backbone. This result revealed the excellent thermal stability of the prepared AEMs, which meet the operating temperature ranges of AEMFCs (generally from 60 to 95 °C).

### 3.4. Ion Conductivity, Alkaline Stability and AEMFCs Performance

The ion conductivity of AEMs is a crucial parameter that affects the AEMFCs performance directly, as high ion conductivity generally contributes to high fuel cell power density. The OH^−^ conductivities of the CPBTT-x AEMs as a function of temperature were measured. As shown in Figure 5a, the OH^−^ conductivities of all the AEMs increases accordingly with the increase of temperature as expected. The CPBTT-0.6 membrane with the highest IEC (1.77 mmol g^−1^) exhibits the highest OH^−^ conductivities of 59.1 mS cm^−1^ at 30 °C and 122.9 mS cm^−1^ at 90 °C, which is competitive to the current AEMs [56,57,58]. Further, the activation energy for OH^−^ conduction in the CPBTT-x AEMs was calculated (Figure 5b). The CPBTT-0.6 exhibits the lowest activation energy of 11.34 kJ mol^−1^ compared to CPBTT-0.4 (13.45 kJ mol^−1^) and CPBTT-0.5 (12.83 kJ mol^−1^), indicating the facilitated OH^−^ kinetic transport. Such a competitive ion conductivity of CPBTT-0.6 is attributed to the well-connected ion transport channels constructed by the hydrogen bonding interaction. 

Apart from ion conductivity, alkaline stability is another important issue for practical AEMFCs’ application. The alkaline durability of the CPBTT-0.6 membrane was investigated by immersing it in 1 M NaOH at 80 °C for 1080 h, and the change in IEC value was recorded during the durability testing. As shown in Figure 5c, CPBTT-0.6 maintains 87.3% IEC after alkaline treatment for 1080 h. Similarly, both CPBTT-0.4 and CPBTT-0.5 are tolerant to harsh alkaline environments, and 86.2 % and 88.7% IEC retention were achieved after the same alkaline tolerance testing, respectively (Appendix A). This result indicates that the CPBTT-x possesses remarkable alkaline stability contributed from the chemically stable aryl ether-free polymer backbone and alkylene chain-modified trimethylammonium group, which is qualified for AEMFCs’ application.

To learn the practical application potential of the prepared AEMs for AEMFCs, the CPBTT-0.6 membrane, which possesses the best comprehensive properties, was chosen for H_2_/O_2_ single fuel cell testing. The AEMFCs performance of CPBTT-0.6 was recorded at 70 °C, 1000/1000 mL min^−1^ H_2_-O_2_ flow rate and 100% RH without back pressure, and the corresponding polarization curve and power density curve were shown in Figure 5d. The AEMFCs containing the CPBTT-0.6 membrane show an open-circuit voltage over 0.94 V, indicating the excellent gas-barrier property of CPBTT-0.6. The CPBTT-0.6 membrane reached a peak power density of 555 mW cm^−2^ at a moderate current density of 1040 mA cm^−2^ without back pressure. The ion conductivity and AEMFCs’ performance of CPBTT-0.6 were compared with recently reported AEMs (Appendix A). Although CPBTT-0.6 possesses competitive ion conductivity, the obtained AEMFCs performance is obviously inferior to the number of AEMs reported currently. This result suggests that the ion conductivity of AEMs may not always be positively correlated to the AEMFCs’ performance, as there are various factors (including MEA fabrication, carbonation of AEMs, etc.) that can affect the AEMFCs’ performance [59]. Future work will be devoted to improving fuel cell performance by optimizing the method for MEA fabrication and testing conditions.

## 4. Conclusions

In summary, a series of novel AEMs with aryl ether-free backbone and urea group-containing ionic side chains have been developed. Experimental studies confirm that the hydrogen bonding interaction between the urea groups facilitates the formation of well-connected ion-conducting channels, which finally benefit the OH^−^ conduction of AEMs. On the other hand, the present AEMs show excellent alkaline stability because of the adoption of a chemically stable ether-free backbone and alkylene chain-modified trimethylammonium cation. Moreover, due to the combination of covalent-bond and hydrogen-bond crosslinking structures, the prepared AEMs show good mechanical robustness and excellent dimensional stability. The single fuel cell test result based on the prepared membrane confirmed the promising application potential of the AEMs. We anticipate that by rational design of the molecular structure, the AEMs’ performance can be improved, thereby improving the performance of the resultant electrochemical devices. Considering the growing demand for AEMs with excellent properties in the industry, we believe our work will provide important insight into the development of high-performance AEMs, and therefore constitute an important step toward AEMs-based electrochemical technologies.

## Figures and Tables

**Figure 1 polymers-14-04920-f001:**
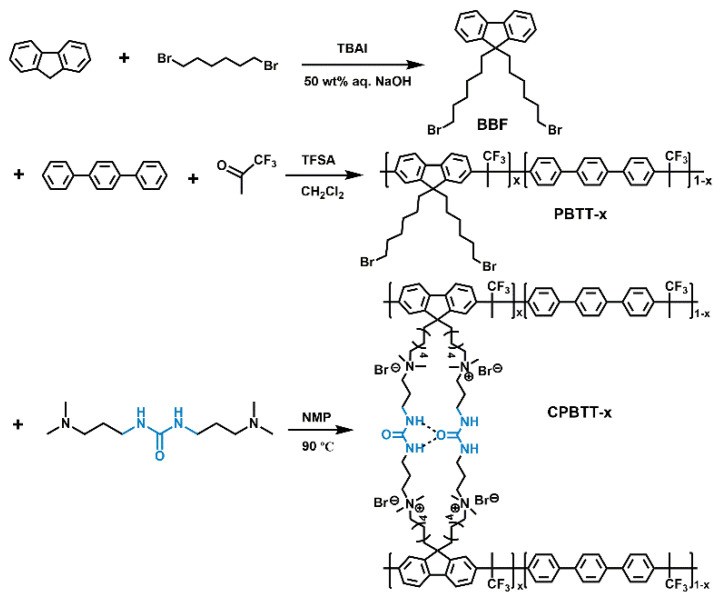
Synthetic route to the CPBTT−x AEMs.

**Figure 2 polymers-14-04920-f002:**
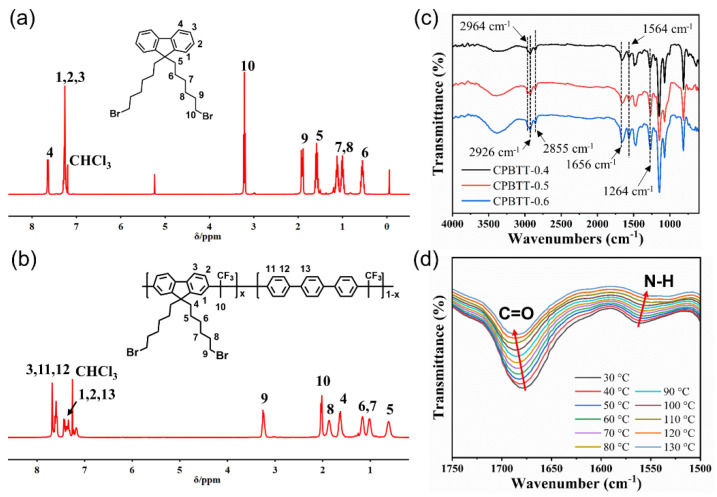
^1^H NMR spectra of (**a**) BBF monomer and (**b**) PBTT−x polymer backbone. (**c**) FTIR spectra of CPBTT−x membranes. (**d**) temperature−dependent FTIR spectra of CPBTT−0.6 membrane.

**Figure 3 polymers-14-04920-f003:**
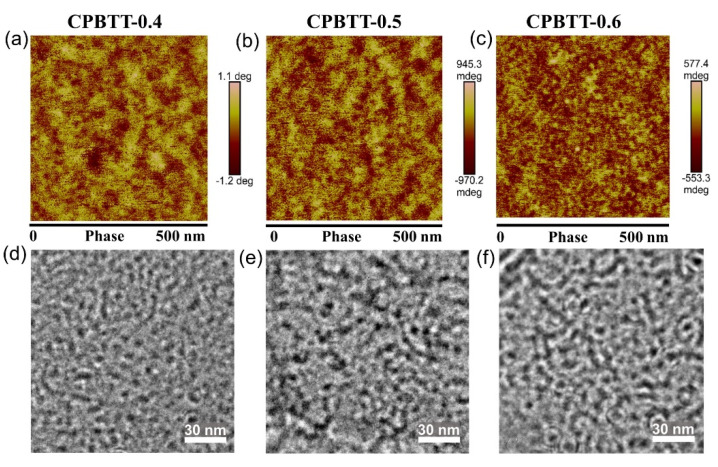
AFM phase images of the (**a**) CPBTT−0.4, (**b**) CPBTT−0.5 and (**c**) CPBTT−0.6 AEMs. TEM images of the (**d**) CPBTT−0.4, (**e**) CPBTT−0.5 and (**f**) CPBTT−0.6 AEMs.

**Figure 4 polymers-14-04920-f004:**
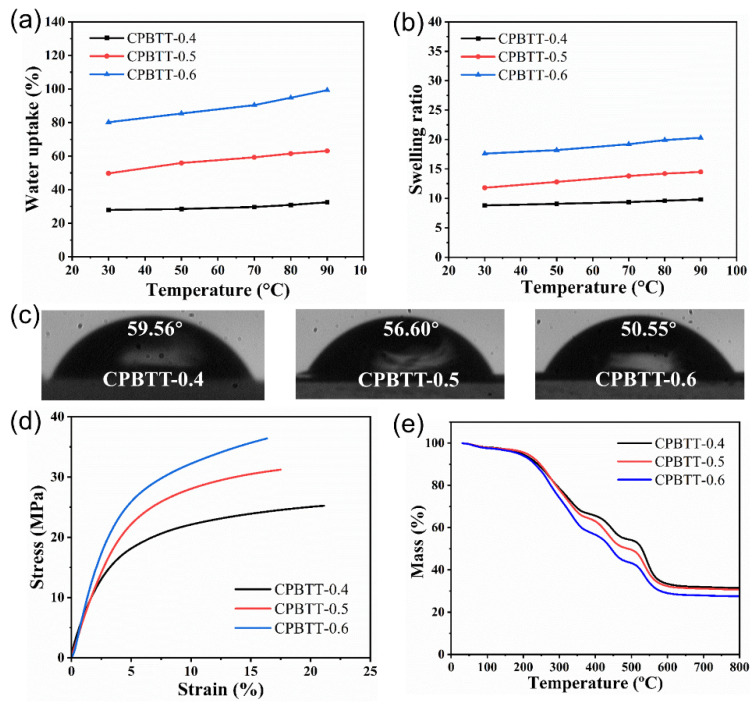
(**a**) Water uptake and (**b**) swelling ratio of the CPBTT−x AEMs at different temperatures. (**c**) the water contact angle of the CPBTT−x AEMs. (**d**) stress−strain curves and (**e**) TGA curves of the CPBTT−x AEMs.

**Figure 5 polymers-14-04920-f005:**
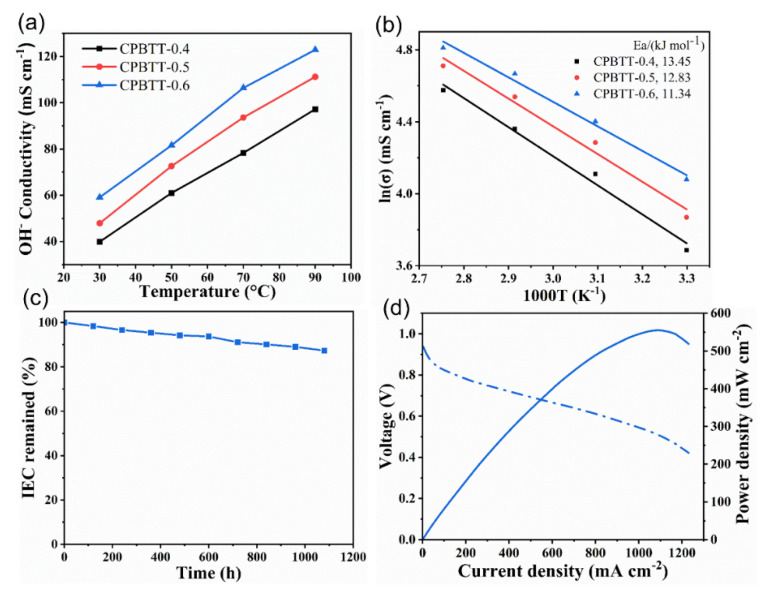
(**a**) OH^−^ conductivity CPBTT−x AEMs as a function of temperature. (**b**) Arrhenius plots of the CPBTT−x AEMs. (**c**) changes in the IEC value of the CPBTT−0.6 AEMs during the durability testing (1 M NaOH at 80 °C for 1080 h). (**d**) the AEMFCs performance of the CPBTT−0.6 AEMs.

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
