# Peer review of "Physically and Chemically Stable Anion Exchange Membranes with Hydrogen-Bond Induced Ion Conducting Channels"

_polymers, 2022, doi:10.3390/polym14224920_

Round 1

Reviewer 1 Report

In my opinion, it is a practical and well written article but it needs to be considered the following comments before publishing.

The title needed significant modification.

The result in abstract part must be extended.

Introduction: They should summarize research findings and novelty. The purpose of the study is not well expressed.

Please use the references for your methods.

FTIR spectra, difficult to follow, provide important peaks

Figure S1; provide more details, magnification, working parameters etc.

Future scope of this study can be added as well as social impact can also be discussed in this paper.

Reviewer 2 Report

This paper describes the synthesis and properties of novel anion exchange membranes (AEMs) with aryl ether-free backbones and urea group-containing ionic side chains. The authors investigated quaternary ammonium-functionalized crosslinked poly(1,6-diphenylpyrene-co-p-terphenyl 1,1,1-trifluoroacetone-co-piperidinium) (CPBTT-x) membranes prepared by the Menshutkin reaction between PBTT-x and 1,3-bis(3(dimethylamino)propyl)urea, where PBTT-x is poly(9,9-bis(6-bromohexyl)fluorene-co-p-terphenyl 1,1,1-trifluoroacetone), x the molar ratio of 9,9-bis(6-bromohexyl)fluorene in the copolymer. Comparative properties of synthesized anion-exchange membranes for Ñ… = 0.4, 0.5, and 0.6 are presented. The properties of theses membranes were studied over a wide temperature range. It has been shown that the hydrogen bonding interactions between the urea groups provide driving force to promote the aggregation of cations, and the resultant well-connected ion-conducting channels benefit the OH− conduction of AEMs. Due to the combine of covalent-bond and hydrogen-bond crosslinking, the designed AEMs show good mechanical robustness and excellent dimensional stability. Various experimental techniques were used, including NMR and FTIR, as well as TEM, AFM and thermogravimetric analysis. In my opinion, this work should be published as is. The only inaccuracies I found in reading the manuscript are related to the spelling of the English words: line 170, “Ploymer” (should be “Polymer”) and line 315 “was chose” (it should be “was chosen”).

Reviewer 3 Report

This paper describes the physically and chemically stable anion exchange membranes engineered by covalent-bond and hydrogen-bond crosslinking. The manuscript is in the scope of the journal. The main suggestions have been given below sequentially:

1) Introduction section: The necessity of the novel physically and chemically stable anion exchange membranes in chemical industry should be better presented.

2) Results and discussion section: Figure 3 should be easier to read. Graphics are too small. In TEM figures the scale is invisible.

3) Results and discussion section: Properties and examined parameters of the tested membranes should be compared with conventional anion-exchange membranes used in another scientific studies. It can be presented in table.

Round 2

Reviewer 3 Report

The authors revised the manuscript taking into account the majority of comments of reviewer.

Author Response

General: The authors revised the manuscript taking into account the majority of comments of reviewer.

Reply: Thank you again for your valuable time on our manuscript. With your help, we are more confident the revised version of our manuscript will be more suitable for publication in the journal.